# Factors Influencing the Technology Adoption Behaviours of Litchi Farmers in China

**Hui Li [1], Diejun Huang [2],\* , Qiuzhuo Ma [3], Wene Qi [1] and Hua Li [1]**

1   College of Economics and Management, South China Agricultural University, Guangzhou 510642, China;
    smarthuili@scau.edu.cn (H.L.); qiwene@scau.edu.cn (W.Q.); fatmartin@scau.edu.cn (H.L.)
2   Institute of Geography and Tourism, Guangdong University of Finance & Economics,
    Guangzhou 510320, China
3   Business School, Guangdong University of Foreign Studies, Guangzhou 510420, China;
    201810036@oamail.gdufs.edu.cn
\*   Correspondence: diejunhuang@gdufe.edu.cn

**Abstract:** Litchi is a traditional tree crop grown in Southern China. Sustainable development of the litchi industry is reliant on technology adoption by farmers. The top grafting technique allows for the introduction of new, quality litchi varieties. The fact that these new varieties ripen earlier or later than the traditional ones helps stabilize litchi prices. Selling new litchi varieties can increase farmers' incomes through higher prices of quality varieties and stabilizing prices by staggering the harvest periods. However, the rate of adoption of top grafting among farmers is low, and up till now, more than half of the litchi trees in China are still traditional litchi varieties. This study explores the factors that influence top grafting adoption by litchi farmers. Using primary data gathered by the China Agriculture Research System of Litchi and Longan (CARSLL) from 567 litchi farming households, a binary logit choice model is employed to determine the factors that influence adoption of litchi top grafting among litchi farmers. The results show that farmers owning larger litchi orchards are more likely to adopt top grafting compared to ones owning smaller orchards. Litchi information accumulation, including experience and training, significantly influences farmers' technology adoption levels. Moreover, a positive attitude toward technology also significantly influences technology adoption behaviours.

**Keywords:** technology adoption; farmer behaviour; binary logit choice model; litchi; top grafting technique; China P.R.

## 1. Introduction

Litchi (lychee) is a tropical fruit native to Southern China. Artificial cultivation of the fruit dates to more than 2000 years ago. According to the China Agriculture Research System of Litchi and Longan (CARSLL), currently, mainland China produces more than half of the world's litchi supply, accounting for over 60% of global litchi production. Litchi's attractive appearance and unique flavour lend it high commercial value. However, all litchis are extremely perishable. Their shelf-lives are only two to four days in ordinary temperatures [1]. It means fresh litchi can only be sold to a relatively small geographic scope. What is more, the harvest period for each litchi variety in a region is extremely concentrated. Therefore, litchi farmers are eager to sell their harvest quickly because mature litchi will perish within a few days, losing all commercial value. Following economic principles, a large quantity of undifferentiated product supplied in a small market at the same time can only result in low prices. In general, litchi bumper harvests cannot correspondingly reward litchi farmers. CARSLL reported that suitable weather in 2018 led to the total litchi production in mainland China increasing

by almost 50% over 2017. However, unfortunately, litchi farmers' total income was even lower than in 2017 because of the collapsed price. Litchi farmers are in the dilemma between quantity and price.

Mixing varieties is a shortcut that helps farmers escape from this dilemma, because the harvest periods are staggered. In addition, new litchi varieties can be sold at much higher prices than traditional varieties in the market. According to CARSLL, the average orchard price of Nuomici (a late maturing, high-quality litchi variety) over the last three years (2016–2018) was three times higher than Heiye (a traditional litchi variety). Furthermore, it is fundamentally practical to adopt the variety changing technique, considering the large amount of traditional litchi varieties in the market. As it is seen in CARSLL, in 2017, two of the three most cultivated litchi varieties were traditional ones (Heiye and Huaizhi), accounting for 45.7% of all litchi orchards in mainland China.

The most popular technique for changing the cultivated litchi variety is called top grafting, which is assessed as the first priority litchi cultivation technique by Delphi method [2]. Compared to other variety control technologies (such as hybridization), top grafting has the advantage of resource saving, thanks to its short recovery period and the low uptake cost. Qin and He conducted an economic benefit evaluation of litchi top grafting in China in a 20 ha orchard, called Xü Qizhi Orchard. Based on the orchard's average sales data from 2000–2004 and 2008–2014, the value of Xü Qizhi Orchard's yield increased 5.71 times after the top grafting technique was applied to change the varieties [3].

As different litchi varieties are ready for harvest at different times, it lengthens the period in which litchi can be supplied to the market within a region. By increasing the varieties cultivated in an orchard, a smoother supply is achieved. A stable litchi supply can lead to a relatively stable high price, and higher quality litchi varieties can also garner higher prices. Therefore, the top grafting technique could greatly increase the incomes of litchi farmers and lead to a sustainable development of the industry.

Even though the top grafting technique can significantly improve litchi farmers' incomes and increase the sustainability of the industry, the accumulated uptake rate for these technologies is not very high (from CARSLL). Increasing the rate at which farmers adopt new techniques has become an even more urgent and important issue than technology innovation in most counties [4]. The aim of this study is to determine the factors that influence the adoption of litchi grafting by litchi farmers, and correspondingly, to find out the ways to improve the adoption rate of litchi grafting so as to sustainably develop the litchi industry.

Previous studies on agriculture technology adoption in China focused mostly on top-down promotion, but relatively less focus on farmer adoption behaviours. Furthermore, the majority of the literature on agriculture technology adoption behaviours worldwide have focused on grain crops (beginning with Bryce's very early research on corn in the U.S. [5]), but the studies specifically focusing on tree crops are rare. Litchi is a typical commercial tree crop, and it has specific characteristics that lead to unique farmer behaviours. For instance, tree crops must be grown for several years before they are ready to harvest. In addition, tree crops generally have a higher, but more fluctuating, market price.

Today's China is an interesting place to study farmer technology adoption behaviours. Firstly, some farming households in China, just as in other developing countries, rely on agriculture for their livelihoods. For these households, agricultural growth can be linked directly to poverty reduction. However, they cannot bear the risks involved in trying new technology [6]. Moreover, for some farming households, just as in other developed countries, most of the younger generation now work in cities, leaving older family members to deal with the land. These older farmers may lack knowledge of and ability to apply new techniques [7]. In comparison, young people who have a proper education and the initiative to work in agriculture are called the New Farmers [8], and one of the characteristics of these New Farmers is their eagerness to adopt new technologies. Lastly, Chinese consumers, and the government, are beginning to put more and more emphasis on high-quality edible agricultural products.

The aim of this study is to provide knowledge on the adoption of litchi top grafting by litchi farmers through identifying the internal and external factors that influence the uptake. The results of

this study can help policy makers understand potential drivers of and barriers to the adoption of top grafting by litchi farmers.

## 2. Methodology and Data

### 2.1. Theoretical Model of Technology Adoption

According to the theory of behavioural economics, farmers facing the decision of whether to adopt a new agricultural technique are assumed to be rational and to pursue maximum self-interest. However, Simon put forward the bounded rationality theory that leads the decision making closer to the real world. The theory of bounded rationality is based on the idea that individuals do not have complete information and have cognitive limits, and thus, individuals are seeking for a 'good enough' decision, which Simon called a satisfactory decision [9].

A farmer's decision to adopt a new technology is affected by the information gained as well as his or her cognitive level and attitude about technology. Based on Rogers' innovation diffusion theory, the 5-step process by which a farmer adopts a new technology is as follows: the farmer gains adoption knowledge, forms an attitude about the technology, makes a decision, implements it, and confirms it [10]. Information on the agriculture technology is diffuse, both formally (such as training) and informally (such as from neighbours). A farmer's gaining of information is affected by his or her ability to access information, such as his or her social capital. A farmer's cognitive level is affected by personal and family characteristics. Additionally, adoption behaviour is both a function of dynamic technology diffusion and a psychological process from cognition to decision, which is affected by many factors. The theory model for a litchi farmer's technology adoption is formulated as follows.

$$
\begin{aligned}
D_i &= f(I_i, C_i, A_i) \\
I_i &= f(B_i, E_i) \\
C_i &= f(P_i, F_i) \\
D_i &= f(B_i, E_i, P_i, F_i, A_i) = \beta_0 + \beta_1 B_i + \beta_2 E_i + \beta_3 P_i + \beta_4 F_i + \beta_5 A_i + \varepsilon
\end{aligned}
\tag{1}
$$

In the formulas, $D$ stands for a farmer's decision regarding technology adoption. $I$ represents the farmer's information gain. $C$ stands for the farmer's cognition level. $A$ represents the farmer's attitude about the technology. $B$ represents the farmer's ability to access information. $E$ represents the farmer's accumulated experience with litchi farming. $P$ represents the farmer's personal characteristics, and $F$ stands for the farmer's family characteristics. $\varepsilon$ is the disturbance term that refers to the uncertainty factors. $\beta_0, \beta_1, \beta_2, \beta_3, \beta_4$, and $\beta_5$ are the impact coefficients, and $i$ is the sample number.

### 2.2. Variable Design

From the theories and literature, the factors influencing farmer technology adoption behaviour could include personal characteristics, family characteristics, technology information accessibility, technology information accumulation, and technology attitude.

As to the personal characteristics, the number of years of formal education and the age have been found to be significant predictors of technology adoption in many studies. These are the most common indicators in innovation adoption research, and it has been found that younger and more educated farmers are more likely to adopt new technologies [11,12].

In terms of family characteristics, many studies show that technology adopters tend to have more land and earn a higher income percentage from that land. This indicates the ability to deal with the farm and a stronger motivation to adopt new techniques [4,13]. However, some studies have found that larger farm sizes and a higher percentage of family income coming from the farm have a negative or insignificant effect on technology adoption. This is normally explained by their risk tolerance level [14].

In terms of technology information accessibility, better access to market and technology information have a positive effect on the technology adoption level [14,15]. Having a relative or friend working

in the market may make it easier for farmers to update price information and to sell out products, and it is particularly important for perishable commercial products. Another information source could be the mass media. Considering that the litchi technology information is a very specific topic, little information on the subject is spread through TV and radio. However, smart mobile phones are widely used in China, so litchi-related information can be gained through search engines. The WeChat application, which is a social app with 10 billion users, is chosen as a variable to represent the ability to access Internet information.

For technology information accumulation, attending a specific training is a direct approach to gaining technology knowledge and reducing subjective uncertainty [16,17]. Years of farming experience is another way to accumulate knowledge, which is to learn by working and gaining information from informal mechanisms, such as other farmers [18]. Farmer experience is normally positively related to technology adoption. However, some studies show that less experienced farmers are more open to new technology, and, hence, are more likely to embrace new techniques [19].

The attitude about technology seems to be a behavioural factor behind technology adoption [20]. When a farmer has a high level of trust in technology, he or she is more likely to adopt new techniques. On the other hand, a low level of trust in technology is a key limitation to technology adoption.

Regional specific contextual factors, such as non-agricultural opportunities, natural agricultural resources, and policy support, will affect farmers' technology-related decisions in each locality [13]. Accordingly, regional factors will influence technology adoption.

### 2.3. Empirical Model

There are only two choices—to use or not use—in most farmers' technology adoption decisions. Our field survey is carried out in 2018 by the CARSLL team. In the questionnaires, responding farmers are asked whether they had used the top grafting technique. The answer to the two options corresponds to the different situation and attributes of each responding farmer. The dependent variable of the technology adoption function is a discrete variable, so a binary discrete choice model is chosen. The binary logit choice model is popular in technology adoption research [21]. Therefore, each farmer's decision regarding adoption of the top grafting technique is represented by a dummy variable as below:

$$D_i = \begin{cases} 1 \text{ } if \text{ } a \text{ } farmer \text{ } adopts \text{ } top \text{ } grafting \\ 0 \text{ } if \text{ } a \text{ } farmer \text{ } does \text{ } not \text{ } adopt \text{ } top \text{ } grafting \end{cases} \tag{2}$$

To quantify the factors influencing farmer decisions on whether to adopt the top grafting technique, the following binary model is constructed based on theories and practices:

$$\ln(\frac{p_i}{1-p_i}) = \alpha + \sum_{n=1}^{n} \beta_n x_{ni} \tag{3}$$

where dependent variable $p_i$ stands for the probability of technology adoption, $\alpha$ stands for the intercept parameter, $\beta$ stands for the vector of regression coefficients, and $x_{ni}$ stands for a vector of $n$ independent variables (such as age and farm size).

### 2.4. Data Collection

#### 2.4.1. Study Area

Litchi is a subtropical to tropical perennial fruit tree that needs an annual average temperature of over 25 °C, over 1200 mm of precipitation, over 1700 h of sunshine, and a short period of low temperatures (about 10 °C) to hasten blossoming. Therefore, the regions for litchi cultivation are limited. According to CARSLL, worldwide, there are 32 countries and regions in which litchi are

commercially cultivated, and the main areas of production are China, India, South Africa, Australia, Mauritius, and Madagascar.

The CARSLL data indicates that there are six provinces in China in which litchi are harvested, and the top two are Guangdong (province) and Guangxi (province). In 2018, the two provinces produced 1,581,600 tonnes and 986,300 tonnes of litchi, respectively, accounting for 52.23% and 32.57% of litchi production in mainland China. Therefore, Guangdong and Guangxi were selected as the study area (Figure 1) to discuss what factors influence a litchi farmer's technology adoption in China.

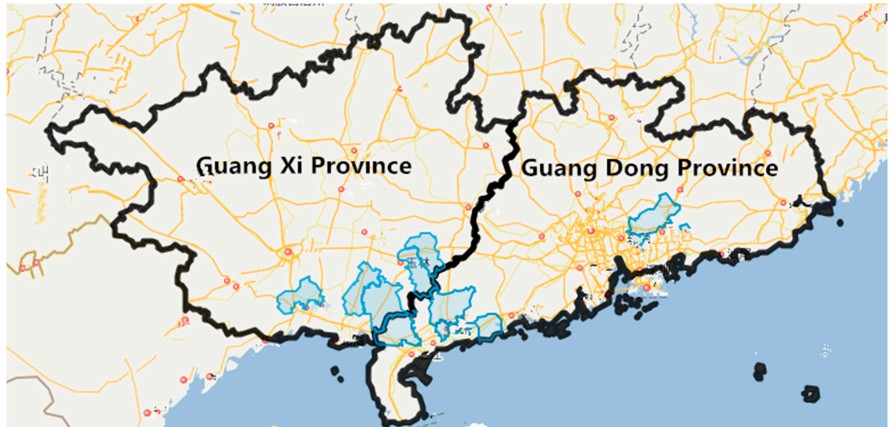

**Figure 1.** Study area.

### 2.4.2. Sampling Method

A farming household survey was conducted by the CARSLL team in July and August of 2018. The sampling method is probability proportionate to size.

First, all 16 Guangdong litchi technology demonstration counties (accounting for 73.74% of all Guangdong litchi planting areas in 2017) were divided by litchi planting size into five groups: large scale, large and medium scale, medium scale, medium and small scale, and small scale. One random sample was picked in each group, resulting in five survey counties. The same sampling method was applied in all 15 Guangxi litchi technology demonstration counties (accounting for 70.06% of all Guangxi litchi planting areas in 2017), so five counties in Guangxi were selected as survey counties. This resulted in a total of 10 survey counties.

Next, all towns in the 10 counties were divided by litchi planting size into three groups: large scale, medium scale, and small scale. One town was randomly sampled in the large-scale and medium-scale groups, respectively, in each country, resulting in 20 survey towns.

After that, all villages in the sample towns were divided by litchi planting size into three groups: large scale, medium scale, and small scale. One village was randomly sampled in the large-scale and medium-scale groups, respectively, in each town, resulting in 40 survey villages. Finally, 25 farmers were randomly sampled from among all the litchi farmers in each survey village, and they were selected as the survey interviewees. This resulted in 1000 farming household survey samples.

To ensure the accuracy and integrity of the survey, all questions in the questionnaire were asked through face-to-face interviews. Similar to the situation in most other rural area in China today, most young and middle-aged farmers were working in towns or cities, leaving old farmers to deal with agriculture, and thus a large number of random sampling samples were not in the village in the survey period. The total number of completed questionnaires was 584, so the response rate was 58.4%.

### 2.4.3. Respondent Profile

Through a data quality audit, the effective rate of the survey was found to be 97.09%. There were 251 questionnaires from Guangdong and 316 from Guangxi. As such, Guangxi accounted for 55.73% of the effective survey samples. Table 1 illustrates the demographic characteristics of the survey samples.

As seen in Table 1, the average age of the respondents was 58.01 years old. Nearly 90% of the farmers were over 45 years old, and 47.90% were over 60 years old. Most of the farmers had a middle school education, accounting for 61.90%, and very few had a higher education (1.77%). 36.33% farmers only had a primary education, and most of them had not even completed that. In total, 39.51% respondents had not received even one training. However, 66.84% of them had more than 20 years of litchi planting experience, and only 11.99% respondents had less than 10 years of experience. The average experience in litchi planting was 22 years. Among the sample, almost two thirds did not use WeChat (63.67%), and 73.02% did not have a relative or friend working in litchi distribution. As to the litchi income as a percentage of total family income, in the largest group (37.57%), litchi income accounted for less than 25% of their total income. However, in the second largest group (31.75%), it accounted for more than 75% of total income. The average litchi orchard size per household was 1.09 ha. However, more than half the farmers (57.32%) had less than 0.7 ha of litchi; 28.21% had 0.7–2 ha, and 6.87% had 2–3.4 ha, with 7.58% farmers having more than 3.4 ha. The largest litchi cultivation area was 23 ha. In terms of attitude about new litchi techniques, a little more than half of the respondents claimed they would apply it when positive results appear, whereas 23.63% claimed that they would use it immediately, and 25.04% would use it when most farmers have adopted it. In all the survey samples, 217 farmers (38.27%) had applied the top grafting technique to their litchi trees, while the other 350 (61.73%) had not.

**Table 1.** Basic sample characters.

| Variables | Value | N = 567 | |
| --- | --- | --- | --- |
| | | Frequency | Percentage (%) |
| Age | <45 | 58 | 10.23 |
| | ≥45 and <60 | 24 | 42.68 |
| | ≥60 | 267 | 47.90 |
| Education level | ≤6 primary school or below | 206 | 36.33 |
| | >6 and ≤12 middle school | 351 | 61.90 |
| | ≥12 college or above | 10 | 1.77 |
| Litchi orchard size | <0.7 ha | 325 | 57.32 |
| | ≥0.7 ha and <2 ha | 160 | 28.21 |
| | ≥2 ha and <3.4 ha | 39 | 6.87 |
| | ≥3.4 ha | 43 | 7.58 |
| Percentage of family income from litchi | <25% | 213 | 37.57 |
| | ≥25 and <50% | 98 | 17.28 |
| | ≥50 and <75% | 76 | 13.40 |
| | ≥75% | 180 | 31.75 |
| Relatives or friends selling litchi | 1 have | 153 | 26.98 |
| | 0 none | 414 | 73.02 |
| Application of WeChat | 1 yes | 206 | 36.33 |
| | 0 no | 361 | 63.67 |
| Litchi training experience | 1 had | 343 | 60.49 |
| | 0 none | 224 | 39.51 |
| Experience of litchi planting | <10 years | 68 | 11.99 |
| | ≥10 and <20 years | 120 | 21.17 |
| | ≥20 years | 379 | 66.84 |
| Attitude towards new technique | 1 apply immediately | 134 | 23.63 |
| | 2 apply when good results appear | 291 | 51.32 |
| | 3 apply when all others applied | 142 | 25.04 |
| Region | 1 Guangdong | 251 | 44.27 |
| | 0 Guangxi | 316 | 55.73 |
| Application of top grafting technique | 1 had applied | 217 | 38.27 |
| | 2 not apply | 350 | 61.73 |

Table 2 presents the survey variable settings and the basic statistics results. Based on the previous literature on farmer behaviour toward agriculture technology adoption, the direction of impact on litchi farmers' decision making in regard to top grafting is predicted.

**Table 2.** Variable settings and descriptions.

| Variable | Variable Definition and Assignment | Average | Standard Deviation | Sign |
|---|---|---|---|---|
| Personal Characteristics | | | | |
| Age | Years of age | 58.01 | 10.72 | – |
| Education level | Years of formal education | 8.20 | 3.10 | + |
| Family characteristics | | | | |
| Litchi orchard size | ha of total litchi orchard | 1.09 | 40.96 | + |
| Proportion of litchi income in total family income | Percentage of total family income from litchi in 2018 | 46.90 | 36.58 | + |
| Ability to access information | | | | |
| Relatives or friends selling litchi | No = 0, Yes = 1 | 0.26 | 0.44 | + |
| Application of WeChat | No = 0, Yes = 1 | 0.36 | 0.48 | – |
| Accumulation of information | | | | |
| Litchi training | No = 0, Yes = 1 | 0.60 | 0.48 | + |
| Years of litchi planting | Years of planting litchi | 21.60 | 9.85 | + |
| Attitude towards technology | | | | |
| Attitude towards new technology | apply immediately = 1, apply when good results appear = 2, apply when most others applied = 3 | 2.08 | 0.38 | – |
| Region | | | | |
| Region | Guangxi = 0, Guangdong = 1 | 0.44 | 0.49 | – |
| Dependent variable | | | | |
| Application of top grafting technique | No = 0, Yes = 1 | 0.38 | 0.48 | |

*2.5. Data Analysis*

2.5.1. Top Grafting Application by Different Types of Farmers

Among all 567 survey samples, 217 had applied the top grafting technique, accounting for 38.27%. According to the different types of farmers and top grafting application, a cross-linked table is generated (Table 3).

**Table 3.** Top grafting technique adoption behaviours of different types of farmers.

| Types of Farmers | | Number | Number of Farmers Applied Top Grafting | Percentage of Farmers Apply Top Grafting Tech (%) |
|---|---|---|---|---|
| Age | <45 | 58 | 18 | 31.03 |
| | ≥45 and <60 | 242 | 97 | 40.08 |
| | ≥60 | 267 | 102 | 38.20 |
| Education level | ≤6 | 206 | 65 | 31.55 |
| | >6 and ≤12 | 351 | 148 | 42.16 |
| | >12 | 10 | 4 | 40.00 |
| Litchi orchard size | <0.7 ha | 325 | 107 | 32.92 |
| | ≥0.7 ha and <2 ha | 160 | 66 | 41.25 |
| | ≥2 ha and <3.4 ha | 39 | 20 | 51.28 |
| | ≥3.4 ha | 43 | 24 | 55.81 |
| Percentage of litchi income in family total | <25% | 213 | 73 | 34.27 |
| | ≥25% and <50% | 98 | 41 | 41.83 |
| | ≥50% and <75% | 76 | 37 | 48.68 |
| | ≥75% | 180 | 66 | 36.66 |
| Relatives or friends selling litchi | Yes | 153 | 66 | 43.14 |
| | No | 414 | 151 | 36.47 |
| Application of WeChat | Apply | 206 | 80 | 38.83 |
| | Not apply | 361 | 134 | 37.11 |
| Litchi training | Had | 343 | 153 | 44.60 |
| | None | 224 | 64 | 28.57 |
| Experience of litchi planting | <10 | 68 | 21 | 30.88 |
| | ≥10 and <20 | 120 | 35 | 29.16 |
| | ≥20 | 379 | 161 | 42.48 |
| Attitude towards new technique | Apply immediately | 134 | 69 | 51.49 |
| | Apply when good results appear | 291 | 98 | 33.67 |
| | Apply when all others applied | 142 | 50 | 35.21 |
| Region | Guangdong | 251 | 59 | 23.50 |
| | Guangxi | 316 | 158 | 50.00 |

The younger age group (<45 years old) had the lowest percentage (31.03%) of applying the top grafting technique. The two other age groups had similar percentages, about 40%. This did not conform to the expectation that younger farmers were more willing to apply the top grafting technique. One possible reason could be that the new generation owns better quality varieties of litchi trees that do not require the top grafting technique. As for the education level, 65 out of the 351 farmers with less than a primary school education (18.52%) had applied the top grafting technique, which was much lower than the average technology adoption level, 38.27%. The other two groups had a similar adoption level; though, the farmers with a college level education had a slightly lower adoption level. One possible reason was that the sample size was too small, as there were only 10 samples. Another possible reason was that their litchi trees might be the better quality varieties.

According to the family characteristics, the group owning larger litchi orchards had a higher top grafting technique application rate. More than half of the farmers had the total litchi orchard size of less than 0.7 ha, but they had the lowest technique adoption rate at 32.92%. The figure for the 0.7–2 ha group was 41.25% and that of the 2–3.4 ha group was 51.28%. The group with more than 3.4 ha was 55.81%. The rate shows a trend that larger farmers are more willing to apply the top grafting technique. Another indicator of family characteristics was the percentage of litchi income in family total income.

The biggest group, whose litchi income was less than 25% of total family income, had the lowest top grafting technique uptake rate. When the percentage of litchi income goes from 25% to 50%, and then to 75%, the technique uptake rate goes from 34.27% to 41.83%, and then to 48.68%. However, when litchi income accounted for more than 75% of total family income, the top grafting uptake rate dropped to 36.66%. The possible reasons for this upside-down U shape curve in the adoption rate are as follows. When litchi income only takes a low percentage of the family income, there is low motivation to adopt new technology. When a higher percentage of a family's income comes from litchi cultivation, there is stronger motivation for farmers to try new techniques to raise their litchi income. However, when litchi income accounts for more than 75% of the total family income, the family has few other income sources besides litchi. This group of farmers are more cautious to try new technology.

Concerning the ability to access information, 26.98% of the farmers had a relative or friend selling litchi, and 36.33% were using WeChat. Farmers who had a relative or friend selling litchi enjoyed a higher percentage of top grafting technique uptake at 43.14%, while the uptake percentage for the others was 36.47%. A person with a job selling litchi is able to gain direct information on the local litchi market. Those that know more about the litchi market are more likely to change from traditional varieties to higher quality varieties. The top grafting rates were very slightly different between the WeChat App group (38.83%) and the non-WeChat group (37.11%). This is possibly because there is still little litchi information available on the Internet.

In terms of information accumulation, 44.60% of farmers who received litchi training had applied the top grafting technique. Correspondingly, 28.57% of farmers who never had litchi training had adopted the technique. The data shows the positive effect of training in technology adoption. Compared with formal training, the number of years of litchi planting experience also showed a positive effect on technology adoption. Farmers with less than 10 years litchi planting experience and 10–20 years of experience had similar top grafting adoption rates—30.88% and 29.16%, respectively. However, 42.48% of farmers with more than 20 years of experience had adopted the top grafting technique.

Attitude to technology is a socio-psychological factor that effects technology adoption behaviour. The farmers, who claimed that they would apply a new technology once they learn it, had the highest top grafting rate among all types of farmers at 51.49%. The application rate of the other groups, including those who claimed they would apply the new technique when it shows good results and those who would apply it only after most others had applied it, were 33.67% and 35.21%, respectively. New technique application requires certain costs and has unpredictable risks. Therefore, the more positive is the farmers' attitude to new techniques, the more possible it would be for them to uptake the top grafting technique.

Different regions have specific external environments that affect farmers. Half of the farmers in Guangxi had adopted the top grafting technique, but the figure in Guangdong was only 23.5%. Comparing the economic development of these two provinces, Guangdong is the number one province in terms of GDP, while Guangxi was number 18 out of all 31 provinces in China. Farmers in Guangdong have, comparatively, more opportunities other than litchi cultivation. On the other hand, Guangxi is an agricultural province, and the fruit industry there has rapidly grown over the past decade. Guangxi and Guangdong are at the same latitude, but in Guangxi, more resources are brought to the agricultural industry, especially for fruits. Data from the Ministry of Agriculture of China showed that, in 2016, the fruit production in Guangxi was 15 million tonnes, and that figure grew to 18 million in 2018.

2.5.2. Factors Influencing a Litchi Farmer's Application of the Top Grafting Technique

A logit analysis was conducted using Stata 12, and the result is shown in Table 4. Based on the field survey samples, the factors positively affecting the application of the top grafting technique include education level, litchi orchard size, and relatives or friends selling litchi. The factors negatively affecting the top grafting technique application include age, percentage of litchi income to family total income, WeChat use, attitude to new technology, and region.

The estimation results of the proportion of litchi income in family total income was the reverse of our expectation. In terms of the different litchi income percentage groups, the rate of adoption of the technique first rose and then fell. The group with a very high percentage (75%) of litchi income in total income did not apply the top grafting technique, and a possible reason is that these families rely so heavily on litchi income that they are unable to bear any risk. Financial support is a must for technology adoption; thus, farmers who are struggling financially would have difficulty with any innovation adoption [21–23].

**Table 4.** Impact of top grafting technique adoption by litchi farmers.

| Variable | Logit Model coef. (Std. Err.) | Marginal Effect coef. (Std. Err.) |
|---|---|---|
| Age | −0.13005 (0.01103) | −0.00036 (0.00228) |
| Education level | 0.06888 (0.03348) | 0.00522 (0.00680) |
| Litchi orchard size | 0.00954 *** (0.00268) | 0.00202 *** (0.00066) |
| Percentage of litchi income to family total income | −0.00007 (0.21017) | −0.00011 (0.00055) |
| Relatives or friends selling litchi | 0.12748 (0.24202) | 0.02295 (0.04266) |
| Application of WeChat | −0.03830 (0.00981) | −0.01743 (0.04951) |
| Litchi training | 0.52518 ** (0.18600) | 0.11205 *** (0.04132) |
| Experience of litchi planting | 0.03553 *** (0.20554) | 0.00678 *** (0.00203) |
| Attitude towards new technology | −0.25101 * (0.00338) | −0.04339 (0.02793) |
| region | −1.19719 *** (0.20192) | −0.24389 *** (0.03634) |
| _cons | −0.86060 (0.91944) | – – |

*, **, *** represent significant difference at the 10%, 5%, and 1% levels, respectively; the values in parentheses are standard deviation.

## 3. Results

The results from the estimated coefficients and marginal effects of the logit model showed that five of the 10 selected variables were significant at the 10% level.

The coefficient of the litchi orchard size variable was positive and statistically significant at the 1% level for farmers adoption of top grafting. Similar results have been found in literature [4,24]; farmers with larger sized farms are more likely to adopt new technology. The reasons for this may be as follows: first of all, the larger farm size itself means more resources. Farmers who own and manage more resources have a higher ability to learn and apply new technologies. Second, a large litchi orchard size could bring in comparatively more income, which can cover the additional operating costs for technology adoption [22]. Third, larger sized farms are more likely to have higher incomes, which motivates farmers to adopt new technology to further improve outcomes.

As to information accumulation, both the litchi training variable and the variable of litchi planting experience had a positive and significant impact on the adoption of the top grafting technique. The litchi training variable showed a 5% significance sign in the adoption equation. This is consistent

with the previous research findings, which have shown that training is an effective tool to improve knowledge, especially in reducing subjective uncertainty about the technology [18,25]. Experience in litchi planting significantly affected top grafting adoption behaviour at the 1% significance level. This is consistent with the notion that farmers with more experience in agriculture are more likely to adopt innovation [26].

The attitude toward technology significantly influenced litchi farmers' adoption of top grafting techniques. The coefficient for technology attitude was negative and significant at the 10% level, suggesting that the litchi farmers who were more eager to apply new technology were more likely to apply the top grafting technique. This result supports the possibility that the attitude toward technology leads to a practical technique uptake [14,27].

The regional factor significantly and negatively affects a litchi farmer's adoption of the top grafting technique at the 1% level. This means that litchi farmers from Guangxi are significantly more likely to apply the top grafting technique than the farmers from Guangdong. The map (Figure 1) shows that the two provinces are adjoined and fall along the same latitude. Both Guangdong and Guangxi are the regions where litchi was originally cultivated, and are still producing the largest percentage of litchi in the world. The difference between the two provinces was that the fruit industry, including litchi, is a more important industry in Guangxi than Guangdong. Guangxi had the second largest fruit production among all 31 Chinese provinces in 2018. A possible reason for this was that the Guangxi government and farmers were more focused on the agriculture industry and paid more attention in technology promotion and adoption.

## 4. Discussion and Conclusions

The large-scale survey on litchi farmers' adoption of technology provides a detailed understanding of common barriers to adoption between adopters and non-adopters. The results of the survey provide political insight and suggest that potential interventions should be encouraged.

In the literature, the main barriers to agriculture technology adoption are cost and risk. This was confirmed in the survey, as larger litchi orchard farmers are more likely to adopt the top grafting technique. Litchi is a kind of tree crop that is perennial for several decades. Smaller sized litchi farms are more likely to belong to, or to have even been passed down from an older generation. These farmers of small sized litchi orchards have no plan to invest in litchi cultivation; they just want to maintain it without incurring any risk. Therefore, as to policy making, litchi technology promotion should target on larger farms that are both willing and able to adopt new technology.

The study results also support the notion that an agriculture extension service is the key to better technology adoption [28]. Most litchi training in China is organized by extension services that show farmers how to apply the technique effectively and reduce uncertainty about the performance. In the survey samples, the average experience of growing litchi was over 20 years, and this group had the highest percentage of farmers accessing extension services. It suggests, in policy making, that agriculture extension services should be promoted to improve litchi technology adoption.

The theory of planned behaviour is built on the assumption that an individual's behaviours are derived from his or her intention, which is primarily determined by attitude [29]. This is also supported by the study, which finds that litchi farmers with a positive attitude toward technology are more likely to adopt new technologies in practice. In terms of policy making, to build up litchi farmers' positive attitude toward technology is a long-term task, which is mainly about promotion and education.

**Author Contributions:** Conceptualization, H.L. (Hui Li) and W.Q.; methodology, Q.M., H.L. (Hui Li), and W.Q.; software, Q.M.; validation, H.L. (Hui Li), D.H., and H.L. (Hua Li); formal analysis, H.L. (Hui Li) and Q.M.; investigation, W.Q., D.H. and X.W.; resources, H.L. (Hui Li) and W.Q.; data curation, D.H.; writing—original draft preparation, H.L. and D.H.; writing—review and editing, H.L. (Hua Li), Q.M., and D.H.; supervision, project acquisition, and administration: H.L. (Hui Li) and D.H. All authors have read and agreed to the published version of the manuscript.

**Funding:** This research was funded by MOE (Ministry of Education in China) Youth Foundation Project of Humanities and Social Sciences (18YJC790079); the earmarked fund for the China Agriculture Research System (CARS-33-18); Natural Science Foundation of Guangdong Province (2018A030310687;2019A1515012149); the Key Project of the National Natural Science Foundation of China (71633002); and Philosophy and Social Science Planning Project of Guangdong Province (GD19CYJ14).

**Conflicts of Interest:** The authors declare no conflict of interest.

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
