# Peer review of "Factors Influencing the Technology Adoption Behaviours of Litchi Farmers in China"

_sustainability, doi:10.3390/su12010271_

Round 1
Reviewer 1 Report
Sustainability – 670537
Title: Factors Influencing the Technology Adoption Behaviours of Litchi Farmers in China
I read this paper with interest. This manuscript determines factors that influence adoption of litchi grafting in Southern China using data collected by China Agriculture Research System of Litchi and Longan (CARSLL) in 2018 from 567 litchi growers in southern China. The authors used a binary logit choice mode to analyze the data and found that area cultivated, experience of litchi farming, training, and positive attitude towards grafting technique affect adoption of litchi grafting by litchi growers in Southern China. The article merits publishing, but it needs a lot of improvement.
In general the paper is poorly written and needs a substantial revision. I recommend sending to a language editor before you submit. The manuscript needs extensive rewriting.
The author could go further and look at the effect of adoption of litchi grafting on income of litchi growers using up-to-date econometric models such as endogenous switching regression, control function, etc. The variables considered for the analysis are also few (they could include more). Just to mention as an example, gender/sex is not considered in the analysis. The manuscript is missing (policy) recommendation(s).
To mention some specific points:
Lines 12-14 (second sentence): too long and lacks clarity
Line 17: change the word ‘technology’ to ‘grafting’; and ‘information’ to ‘data’
Line 18: change the word ‘for’ to ‘from’; and ‘farmer’ to ‘farming’
Line 19/20: change ‘a binary logit choice model is employed to determine the factors influencing litchi farmers’ technology adoption behaviours’ to ‘a binary logit choice model is employed to determine the factors that influence adoption of litchi grafting among litchi growers’.
Line 20/21: change ‘The study results show that farmers that cultivate larger litchi orchards are more willing to adopt new techniques’ to ‘The results show that farmers owning larger litchi orchards are more likely to adopt grafting compared to ………..’.
Line 29: It reads ‘Data from the China….’
You cannot use data as a reference.
Line 36: change ‘spoil’ to ‘perish’
Line 54: delete ‘of the technology’
Line 65: delete ‘technology, such as’
Line 67: ‘(CARSLL data)’ It is not possible to cite data.
Line 69/71: change ‘The aim of this study is to determine the factors influencing litchi farmers’ technology adoption and, correspondingly, how to improve the technology adoption rate to sustainably develop the litchi industry’ to ‘The aim of this study is to determine the factors that influence adoption of litchi grafting by litchi farmers, and correspondingly, how to improve the adoption rate of litchi grafting to sustainably develop the litchi industry’.
Line 91/92: change ‘The aim of this study is to provide an understanding of litchi farmers’ technology adoption behaviours and to identify the internal and external factors influencing their technology uptake’ to ‘The aim of this study is to provide knowledge on adoption of litchi grafting by litchi growers by identify internal and external factors influencing uptake’.
Line 93/94: change ‘The results of this study can help policy makers to understand potential drivers of and barriers to technology adoption by litchi farmers.’ to ‘The results of this study can help policy makers to understand potential drivers of and barriers to the adoption of grafting by litchi growers.’
Line 137: change ‘mess’ to ‘mass’
Equation 2: change ‘frafting’ to ‘grafting’
Line 180: change ‘output’ to ‘production’
Line 185: delete ‘in’ and put ‘,’ after August
Line 86: delete ‘Prof. Wene Qi is the survey general leader’
Line 187/190: is a repetition
Lines 191/201: the classifications into large, medium, etc lack clarity
Lines 206/208: the response rate (58.4) is very low. This means that the sample farms are not anymore representative.
Line 210/231: Table 1 is not mentioned in the text. The text is a direct copy of the table. Please write only the main ones.
You use ‘mu’ as unit of measurement. Is it IS unit. You can use only IS units of measurement in manuscripts.
Table 2: change ‘impact expectations’ to ‘sign’
Line 249: ‘farmers with a university level education had a slightly lower adoption level compare to ….’
Line 258: ‘The biggest group …’. It is not clear
Line 267/268: ‘This farmer group has a low risk tolerance and is not willing to try new technology’. This argument is not convincing unless you give more details about the group. Also change ‘This farmer group’ to ‘This group of farmers’
Line 279: change ‘affect’ to ‘effect’
Table 3: change ‘No of farmers apply top grafting tech’ to ‘Number of farmers applied top grafting’
Line 306: Please be consistent in the use of words and phrases. For instance ‘litchi planting size’ what you are referring to is not clear.
Table 4. Too many numbers after decimal places and no ‘0’ before decimal
Line 323: ‘marginal effect’. There is no result of marginal effect analysis anywhere in the manuscript.
Author Response
Response to Reviewer 1 Comments
Thanks very much for your comments.
Board comments
The author could go further and look at the effect of adoption of litchi grafting on income of litchi growers using up-to-date econometric models such as endogenous switching regression, control function, etc. The variables considered for the analysis are also few (they could include more). The manuscript is missing (policy) recommendation(s).
Response:
Our study is a part of national level litchi survey (the China Agriculture Research System of Litchi and Longan, CARSLL) including a lot of biological questions. Limited by the length of the questionnaire, we cannot set more questions to analyze more variables. Our study focuses on litchi farmers, who are little concern in previous literature. The aim of this study is to find out some general significance in litchi farmers’ technology adoption behaviours. Our econometric model is simple but can meet the need of the study aim. Thanks for your suggestion and we will consider up-to-date models in future in-deep study. The recommendations are in Lines 379-371, Lines 376-377 and Lines 381-383. (in yellow shading)
Specific comments:
Point 1: Lines 12-14 (second sentence): too long and lacks clarity
Response 1: we had revised the long sentence to several short clear ones. (Lines 12-16)
‘The top grafting technique allows for the introduction of new, quality litchi varieties. The fact that these new varieties ripen earlier or later than the traditional ones helps stabilize litchi prices. Selling new litchi varieties can increase farmers’ incomes through higher-prices of quality varieties and stabilizing prices by staggering the harvest periods.’
Point 2: Line 17: change the word ‘technology’ to ‘grafting’; and ‘information’ to ‘data
Response 2: we had revised as your suggestion. (Line 18)
Point 3: Line 18: change the word ‘for’ to ‘from’; and ‘farmer’ to ‘farming’
Response 3: we had revised the phase as your suggestion in this line and elsewhere in the manuscript. (Line 19, etc)
Point 4: Line 19/20: change ‘a binary logit choice model is employed to determine the factors influencing litchi farmers’ technology adoption behaviours’ to ‘a binary logit choice model is employed to determine the factors that influence adoption of litchi grafting among litchi growers
Response 4: we had revised as your suggestion and elsewhere in the manuscript. (Lines 20-21, etc)
Point 5: Line 20/21: change ‘The study results show that farmers that cultivate larger litchi orchards are more willing to adopt new techniques’ to ‘The results show that farmers owning larger litchi orchards are more likely to adopt grafting compared to ………..’.
Response 5: we had revised as your suggestion in these lines and elsewhere in the manuscript. (Lines 21-22, etc)
Point 6: Line 29: It reads ‘Data from the China….’ You cannot use data as a reference
Response 6: we had revised as your suggestion. (Line 31)
Point 7: Line 36: change ‘spoil’ to ‘perish’
Response 7: we had revised as your suggestion. (Line 38)
Point 8: Line 54: delete ‘of the technology’
Response 8: we had revised as your suggestion. (Line 56)
Point 9: Line 65: delete ‘technology, such as’
Response 9: we had revised as your suggestion. (Line 67)
Point 10: Line 67: ‘(CARSLL data)’ It is not possible to cite data.
Response 10: we had revised as your suggestion. (Line 69)
Point 11: Line 69/71: change ‘The aim of this study is to determine the factors influencing litchi farmers’ technology adoption and, correspondingly, how to improve the technology adoption rate to sustainably develop the litchi industry’ to ‘The aim of this study is to determine the factors that influence adoption of litchi grafting by litchi farmers, and correspondingly, how to improve the adoption rate of litchi grafting to sustainably develop the litchi industry’.
Response 11: we had revised as your suggestion. (Lines 70-73)
Point 12: Line 91/92: change ‘The aim of this study is to provide an understanding of litchi farmers’ technology adoption behaviours and to identify the internal and external factors influencing their technology uptake’ to ‘The aim of this study is to provide knowledge on adoption of litchi grafting by litchi growers by identify internal and external factors influencing uptake’.
Response 12: we had revised as your suggestion. (Lines 93-94)
Point 13: Line 93/94: change ‘The results of this study can help policy makers to understand potential drivers of and barriers to technology adoption by litchi farmers.’ to ‘The results of this study can help policy makers to understand potential drivers of and barriers to the adoption of grafting by litchi growers.’
Response 13: we had revised as your suggestion. (Lines 94-96)
Point 14: Line 137: change ‘mess’ to ‘mass’
Response 14: we had revised as your suggestion. (Line 142)
Point 15: Equation 2: change ‘frafting’ to ‘grafting
Response 15: we had revised as your suggestion. (Equation 2)
Point 16: Line 180: change ‘output’ to ‘production’
Response 16: we had revised as your suggestion in this line and elsewhere in the manuscript. (Line 181, etc)
Point 17: Line 185: delete ‘in’ and put ‘,’ after August
Response 17: we had revised as your suggestion. (Line 192)
Point 18: Line 186: delete ‘Prof. Wene Qi is the survey general leader
Response 18: we had revised as your suggestion. (Line 193)
Point 19: Line 187/190: is a repetition
Response 19: we deleted as your suggestion. (Line 193)
Point 20: Lines 191/201: the classifications into large, medium, etc lack clarity
Response 20: the sampling method is probability proportionate to size, which in practise has a complex grouping process. There are 5 groups: large/ large and medium/ medium/ medium and small/ small. We tried to revise a bit to make it clear. (Lines 194-200)
Point 21: Lines 206/208: the response rate (58.4) is very low. This means that the sample farms are not anymore representative.
Response 21: we gave some explanation to the low response rate.
‘Similar to the situation in most other rural area in China today, most young and middle-aged farmers were working in towns or cities while leaving old farmers to deal with agriculture, and thus a large number of random sampling samples were not in the village in the survey period.’
Point 22: Line 210/231: Table 1 is not mentioned in the text. The text is a direct copy of the table. Please write only the main ones
Response 22: we had revised as your suggestion to delete some part of the text. (Lines 217-237) Table 1 is mentioned in Line 219 and Line 220.
Point 23: You use ‘mu’ as unit of measurement. Is it IS unit. You can use only IS units of measurement in manuscripts.
Response 23: “mu” is a Chinese local measurement. We had revised as your suggestion to use “ha” instead through the manuscript. (Line 57, etc)
Point 24: Table 2: change ‘impact expectations’ to ‘sign
Response 24: we had revised as your suggestion. (Table 2)
Point 25: Line 249: ‘farmers with a university level education had a slightly lower adoption level compare to ….’
Response 25: we had revised as your suggestion to make it clearer. (Lines 254-255)
‘The other two groups had a similar adoption level, though, the farmers with a college level education had a slightly lower adoption level.’
Point 26: Line 258: ‘The biggest group …’. It is not clear
Response 26: we had revised as your suggestion to make it clearer. (Lines 264-265)
‘The biggest group, whose litchi income was less than 25% of total family income, had the lowest top grafting technique uptake rate.’
Point 27: Line 267/268: ‘This farmer group has a low risk tolerance and is not willing to try new technology’. This argument is not convincing unless you give more details about the group. Also change ‘This farmer group’ to ‘This group of farmers’
Response 27: we had revised it to ‘However, when litchi income accounts for more than 75% of the total family income, the family has few other income sources, but litchi. This group of farmers are more cautious to try new technology. (Lines 272-274)
Point 28: Line 279: change ‘affect’ to ‘effect’
Response 28: we had revised as your suggestion. (Line 285)
Point 29: Table 3: change ‘No of farmers apply top grafting tech’ to ‘Number of farmers applied top grafting’
Response 29: we had revised as your suggestion in this line and all other similar situations in the manuscript. (Table 3)
Point 30: Line 306: Please be consistent in the use of words and phrases. For instance ‘litchi planting size’ what you are referring to is not clear.
Response 30: we had revised as your suggestion. When it is about one farmer’s growing size, we used ‘litchi orchard size’. When it is about a region’s total litchi planting size, we use ‘litchi planting size’. (Line 22, etc)
Point 31: Table 4. Too many numbers after decimal places and no ‘0’ before decimal
Response 31: we had revised as your suggestion. (Table 4)
Point 32: Line 323: ‘marginal effect’. There is no result of marginal effect analysis anywhere in the manuscript
Response 32: we had revised as your suggestion. (Table 4)

Reviewer 2 Report
The manuscript has scientific novelty and is presented with due quality. The aim of this manuscript is to provide an understanding of litchi farmers’ technology adoption behaviors and to identify the internal and external factors influencing their technology uptake. This is a very relevant and interesting topic, in connection to China’s important production volume (ca. 60% of world production), how perishable the litchi fruit is, and the role technology adoption could play in future. This is in my perception the main addition of the manuscript to the subject area, and its novelty. The methodology or overall research approach of this manuscript are not necessarily part of its novelty. The manuscript allows a smooth reading. It is clear and overall well written, however minor English-language improvements shall be made. The conclusions address the main question posed, and are consistent with the evidence and arguments. The manuscript concludes that larger litchi farmers tend to be more likely to adopt the top grafting technique, and that an agriculture extension service is key for technology adoption. If it would still be possible to pose a question to the authors, I would kindly ask them to highlight what is in their perception the novelty of the manuscript. In summary, in my opinion this is a manuscript that deserves to be published in this journal due to the relevance of the topic it addresses (China’s litchi production and technology adoption), however the manuscript is not scientifically innovative in its approach or methodology.
My only suggestion would be:
Unless otherwise advised by the journal guidelines, consider a different set of keywords, not so similar to the words in the title of the manuscript.
Author Response
Response to Reviewer 2 Comments
Thanks very much for your comments.
Board comments
The manuscript has scientific novelty and is presented with due quality. The aim of this manuscript is to provide an understanding of litchi farmers’ technology adoption behaviors and to identify the internal and external factors influencing their technology uptake. This is a very relevant and interesting topic, in connection to China’s important production volume (ca. 60% of world production), how perishable the litchi fruit is, and the role technology adoption could play in future. This is in my perception the main addition of the manuscript to the subject area, and its novelty. The methodology or overall research approach of this manuscript are not necessarily part of its novelty. The manuscript allows a smooth reading. It is clear and overall well written, however minor English-language improvements shall be made. The conclusions address the main question posed, and are consistent with the evidence and arguments. The manuscript concludes that larger litchi farmers tend to be more likely to adopt the top grafting technique, and that an agriculture extension service is key for technology adoption. If it would still be possible to pose a question to the authors, I would kindly ask them to highlight what is in their perception the novelty of the manuscript. In summary, in my opinion this is a manuscript that deserves to be published in this journal due to the relevance of the topic it addresses (China’s litchi production and technology adoption), however the manuscript is not scientifically innovative in its approach or methodology.
Response: Our study focuses on litchi farmers, who are little concern in previous literature. The aim of this study is to find out some general significance in litchi farmers’ technology adoption behaviours. Our econometric model is lack of novelty but can meet the need of the study aim. Thanks for your suggestion and we will consider more innovative approach in future in-deep study.
Specific comments:
Unless otherwise advised by the journal guidelines, consider a different set of keywords, not so similar to the words in the title of the manuscript.
Response: we added ‘top grafting technique’, ‘binary logit choice model’ as keywords.

Round 2
Reviewer 1 Report
Line 55: insert 'grown' after crop
line 56: replace 'upon' by on